# Comparative effectiveness of ultrasound-guided and anatomic landmark percutaneous dilatational tracheostomy: A systematic review and meta-analysis

**Kun-Te Lin**[1], **Yung-Shuo Kao**[2], **Chun-Wen Chiu**[1,3,4], **Chi-Hsien Lin**[1], **Chu-Chung Chou**[1,5,6,7], **Pei-You Hsieh**[1], **Yan-Ren Lin**[1,5,6,7]*

1 Department of Emergency and Critical Care Medicine, Changhua Christian Hospital, Changhua, Taiwan, 2 Department of Radiation Oncology, China Medical University Hospital, Taichung, Taiwan, 3 Department of Education, National Chiayi University, Chiayi, Taiwan, 4 Department of Nursing, Da-Yeh University, Changhua, Taiwan, 5 School of Medicine, Kaohsiung Medical University, Kaohsiung, Taiwan, 6 School of Medicine, Chung Shan Medical University, Taichung, Taiwan, 7 College of Medicine, National Chung Hsing University, Taichung, Taiwan

* h6213.lac@gmail.com

**Data Availability Statement:** All relevant data are within the manuscript.

## Abstract

### Introduction

Ultrasound-guided tracheostomy (UGT) and bronchoscope-guided tracheostomy (BGT) have been well compared. However, the differences in benefits between UGT and landmark tracheostomy (LT) have not been addressed and, in particular, lack a detailed meta-analysis. We aimed to compare the first-pass success, complication rate, major bleeding rate, and tracheostomy procedure time between UGT and LT.

### Methods

In a systematic review, relevant databases were searched for studies comparing UGT with LT in intubated patients. The primary outcome was the odds ratio (OR) of first-pass success. The secondary outcomes were the OR of complications, OR of major bleeding, and standardized mean difference (SMD) of the total tracheostomy procedure time.

### Results

The meta-analysis included three randomized controlled studies (RCTs) and one nonrandomized controlled study (NRS), comprising 474 patients in total. Compared with LT, UGT increased first-pass success (OR: 4.287; 95% confidence interval [CI]: 2.308 to 7.964) and decreased complications (OR: 0.422; 95% CI: 0.249 to 0.718). However, compared with LT, UGT did not significantly reduce major bleeding (OR: 0.374; 95% CI: 0.112 to 1.251) or the total tracheostomy placement time (SMD: -0.335; 95% CI: -0.842 to 0.172).

**Funding:** The authors received no specific funding for this work.

**Competing interests:** The authors have declared that no competing interests exist.

## Conclusions

Compared with LT, real-time UGT increases first-pass success and decreases complications. However, UGT was not associated with a significant reduction in the major bleeding rate. The total tracheostomy placement time comparison between UGI and LT was inconclusive.

## Introduction

Compared with surgical tracheostomy, percutaneous dilatational tracheostomy (PDT) requires less operation time [1] and has a similar risk of procedure-related complications [1, 2]. Moreover, PDT offers potential benefits such as increased patient comfort, decreased sedation requirements, and decreased dead space [3]. Clinically, PDT can be classified as ultrasound-guided percutaneous tracheostomy (UGT), bronchoscope-guided percutaneous tracheostomy (BGT), and traditional anatomic landmark percutaneous tracheostomy (LT) methods [1].

BGT provides benefits in the real-time confirmation of needle placement, midline positioning of the needle, and avoidance of posterior tracheal wall injury [4, 5]. However, the benefit of reducing the complication rate is not significantly observed [6, 7]. A previous study also noted that increased airway resistance during BGT would secondary increase high airway pressure [8]. Therefore, considering patient safety and cost effectiveness, BGT is challenging among PDT procedures [7].

The other two methods, UGT and LT, have been promoted in different studies. LT is less expensive and requires less operation time than BGT [8, 9]. LT is easily accessible even if hospitals do not have sufficient resources. More importantly, compared with BGT, several studies have demonstrated that the risk of procedure-related complications is not higher in LT [7, 10]. In contrast, UGT is an operator-dependent, noninvasive, and real-time image-guided operation. UGT can identify tube insertion sites during the PDT procedure [5, 11]. Moreover, it can also identify anatomic structures and avoid vessel punctures [12, 13]. UGT and BGT have been well compared [14–20]. However, the comparison between UGT and LT has not been well addressed and especially lacks a detailed meta-analysis. Therefore, in this study, we aimed to compare the first-pass success rate, complications, major bleeding, and tracheostomy procedure time between UGT and LT.

## Methods

### Search strategy and inclusion criteria

Electronic searches were performed using PubMed, Embase, the Cochrane Central Register of Controlled Trials and the Cochrane Database of Systemic Reviews through January 31, 2021. Studies reporting comparisons between UGT and LT in intubated patients were included. We excluded studies with pediatric patients, studies without real-time ultrasound guidance during the UGT procedure, and studies using the surgical tracheostomy method. The research question was defined by the PICO model in accordance with the Preferred Reporting Items for Systematic Reviews and Meta-Analyses (PRISMA) statement [21] (Population: patient intubated with ventilator support; Intervention: real-time UGT; Comparison: LT; Outcome: complication rate, first-pass success rate, major bleeding rate, and tracheostomy procedure time. The search terms were "percutaneous tracheostomy" AND "ultrasound" OR "echography". The

bibliographies of the included studies and related review articles were reviewed for references. Literature not written in English or not available in full text was excluded.

We enrolled randomized controlled trials (RCTs) and non-RCTs, and case series studies or case reports were excluded. All studies included two interventional modalities: real-time UGT and LT.

## Data extraction and quality assessment

All retrieved articles and data were reviewed by two reviewers. We recorded the first author, year, number of patients, number of analyzed patients, study design method, average age, number of males and females, patient characteristics, and intervention details. Two reviewers evaluated the involved studies by means of Jadad scoring for RCTs and the Newcastle-Ottawa Quality Assessment Scale for nonrandomized controlled studies (NRSs). The Jadad score evaluates RCTs according to three aspects: randomization (2 points), blinding (2 points), and an account of all patients (1 point). A higher score indicates better methodological quality. The Newcastle-Ottawa Quality Assessment Scale contains nine items in three categories: participant selection (four items), comparability (two items), and exposure (three items). Disagreement between reviewers was resolved by discussion.

Two reviewers assessed the risk of bias according to the Cochrane Handbook for Systematic Reviews of Interventions guidelines [22]. Seven domains were assessed for each study: random sequence generation, allocation concealment, blinding of participants and personnel, blinding of outcome assessment, incomplete outcome, selection bias, and other bias. Each domain was regarded as low risk (green), unclear risk (yellow), or high risk (red).

## Data synthesis and analysis

The odds ratios (ORs) of first-pass success between UGT and LT comprised the primary outcome. The ORs of complications, ORs of major bleeding and standardized mean difference (SMD) of total tracheostomy placement time between UGT and LT were secondary outcomes. Total tracheostomy placement time was measured in minutes, from the beginning of needle puncture to the completion of tracheostomy tube placement. A random effect model was used to pool the SMDs and ORs individually. The pooled ORs and SMDs were shown in forest plot. The results were analyzed by Comprehensive Meta-Analysis (CMA) Software, version 3 (Biostat, Englewood, NJ, USA). Between-trial heterogeneity was determined by $I^2$ tests; values > 50% were regarded as considerable heterogeneity. Funnel plots, analyzed by R language using R studio, were used to examine potential publication bias. Statistical significance was defined as p-values < 0.05, except for publication bias, which employed p < 0.10.

## Results

### Literature review

The results of the literature review are shown in Fig 1. A total of 282 publications were identified by scanning the databases. After duplicate removal, 234 publications were selected for meticulous evaluation. After assessment of the titles and abstracts, five publications were selected, and their full texts were retrieved. One short communication of RCTs was excluded because the included experimental process was not clear [23]. Finally, four publications, including three RCTs [24–26] and one NRS [27], were selected for the meta-analysis.

The selected studies comprised 474 patients, 242 with the UGT method and 232 with the LT method. All patients were in intensive care units (ICUs) and were intubated with invasive ventilator support. One study compared three different percutaneous tracheostomy methods:

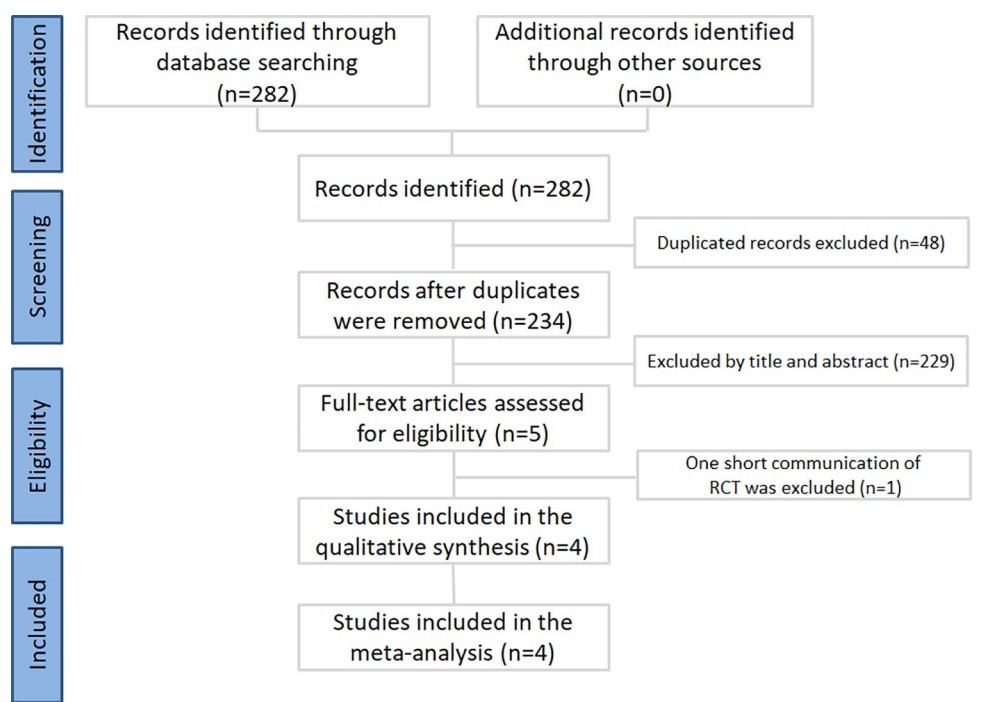

**Fig 1. Inclusion process for the identified studies.**

real-time long-axis, real-time short-axis, and traditional landmark techniques [26]. One study compared only the real-time long-axis UGT method with the LT method [27]. The Jadad scores of the included RCTs were between two and four points. The randomization and blinding methods were recorded in Rudas's study [24], but they were not recorded in the other two RCTs [25, 26]. In Dinh's study [27], the Newcastle-Ottawa Quality Assessment scale was eight points, and the study controls for additional factors were not clearly reported. The patient characteristics and study methodology of the included trials are listed in Tables 1 and 2, while Table 3 summarizes the UGT and LT procedures.

**Table 1. Characteristics of included studies comparing ultrasound-guided with landmark percutaneous dilatational tracheostomy.**

| Author, year | Number of patients | Number of patients analyzed | | Study design | Average age male/female | | Population | Quality assessment |
|---|---|---|---|---|---|---|---|---|
| | | Ultrasound guided | Landmark | | Ultrasound guided | Landmark | | |
| Rudas et al., 2014 | 50 | 23 | 24 | RCT | 57.0±15.1 | 58.4±15.2 | ICU patients | 4* |
| | | | | | M/F: (12/13) | M/F: (12/13) | | |
| Yavuz et al., 2014 | 341 | 154 | 167 | RCT | 59.6±14.9 | 57.5±11.3 | ICU patients | 2* |
| | | | | | M/F: (96/70) | M/F: (99/76) | | |
| Kupeli I, 2017 | 60 | Long axis: 20 | 20 | RCT | Long axis: 64.2±17.4 | 71.0±12.5 | ICU patients | 2* |
| | | | | | M/F: (12/8) | M/F: (12/8) | | |
| | | Short axis: 20 | | | Short axis: | | | |
| | | | | | M/F: (13/7) | | | |
| Dinh et al., 2014 | 23 | 11 | 12 | NRS | 56±18 | 50±21 | ICU patients | 8# |
| | | | | | M/F: (6/5) | M/F: (6/6) | | |

Note: RCT: Randomized controlled trial; NRS: Nonrandomized controlled study; ICU: Intensive care unit; * indicates that the study was evaluated by the Jadad score
# indicates that the study was evaluated by the Newcastle-Ottawa Quality Assessment Scale

**Table 2. Jadad score assessment of RCTs and Newcastle-Ottawa Quality Assessment Scale of NRS.**

| | Jadad score | | | | |
|---|---|---|---|---|---|
| Author, year | Randomization | | Blinding | | Withdrawals and dropouts |
| | | Appropriate method mentioned | | Appropriate method mentioned | |
| Rudas et al., 2014 | ★ | ★ | ★ | ☆ | ★ |
| Yavuz et al., 2014 | ★ | ☆ | ☆ | ☆ | ★ |
| Kupeli I, 2017 | ★ | ☆ | ☆ | ☆ | ★ |

| | Newcastle-Ottawa Quality Assessment Scale | | | | | | | |
|---|---|---|---|---|---|---|---|---|
| Author, year | Adequate case definition | Representativeness of the cases | Selection of controls | Definition of controls | Comparability of cases and controls | Assessment of outcome | Follow-up long enough | Adequate follow up of controls |
| Dinh et al., 2014 | ★ | ★ | ★ | ★ | ★☆ | ★ | ★ | ★ |

★ indicates fulfilment of the criteria; ☆ indicates no fulfilment of the criteria

## Risks of bias in the included studies

The risk of selection bias in the included studies was low, except for one NRS, which had high selection bias. The blinding of participants and personnel was high due to the clinical setting and the process of the procedures. The blinding of outcome assessment was not recorded in most included studies. The reporting bias was low, and reporting data were complete in the included studies. The other risk of bias in the included studies is shown in Fig 2.

**Table 3. Summary of the intervention details in the included studies.**

| Author, year | Interventions | | Outcome measurement |
|---|---|---|---|
| | Ultrasound guided | Landmark | |
| Rudas et al., 2014 | Identification of the midline trachea and cricoid ring with the ultrasound longitudinal probe position. Real-time trachea puncture with the out-of-plane technique, between the first and fourth trachea rings. Tracheostomy tube insertion after guidewire insertion and dilation. | Palpation of the anatomical landmarks to locate the puncture site. | First-pass success |
| | | | Complications |
| Yavuz et al., 2014 | A radiologist performed a real-time ultrasound-guided trachea puncture through the midline level and introduced the guidewire. Another clinician completed the dilation and tracheostomy placement. | Physical landmarks were used to locate the puncture site. | Total tracheostomy procedure time (min) |
| | | | First-pass success |
| | | | Complications |
| Kupeli I, 2017 | Long axis: Real-time ultrasound-guided in-plane method. Needle puncture between the first and second trachea rings. | The space between first and second, or second and third trachea rings was selected by palpation. The inserted puncture needle was directed caudally. Tracheostomy tube insertion was performed after guidewire insertion and dilation. | Total tracheostomy procedure time (min) |
| | Tracheostomy tube insertion after guidewire insertion and dilation. | | First-pass success |
| | Short axis: Real-time ultrasound-guided out-of-plane method. Needle puncture between the first and second or the second and third trachea rings. | | Complications |
| | Tracheostomy tube insertion after guidewire insertion and dilation. | | |
| Dinh et al., 2014 | Obtain the midline position by the ultrasound short-axis technique. Under real-time sonographic guidance in the long-axis view, the needle is inserted into the first or second trachea space, and the guidewire is passed. Place a Shiley number 8 tube after dilation. | Identify the midline trachea by palpation, and make a 1 to 1.5 cm vertical incision inferior to the cricoid cartilage. Select a tracheal ring, introduce the needle, and pass the guidewire. | Total tracheostomy procedure time (min) |
| | | | First-pass success |
| | | Place a Shiley number 8 tube after dilation. | Complications |

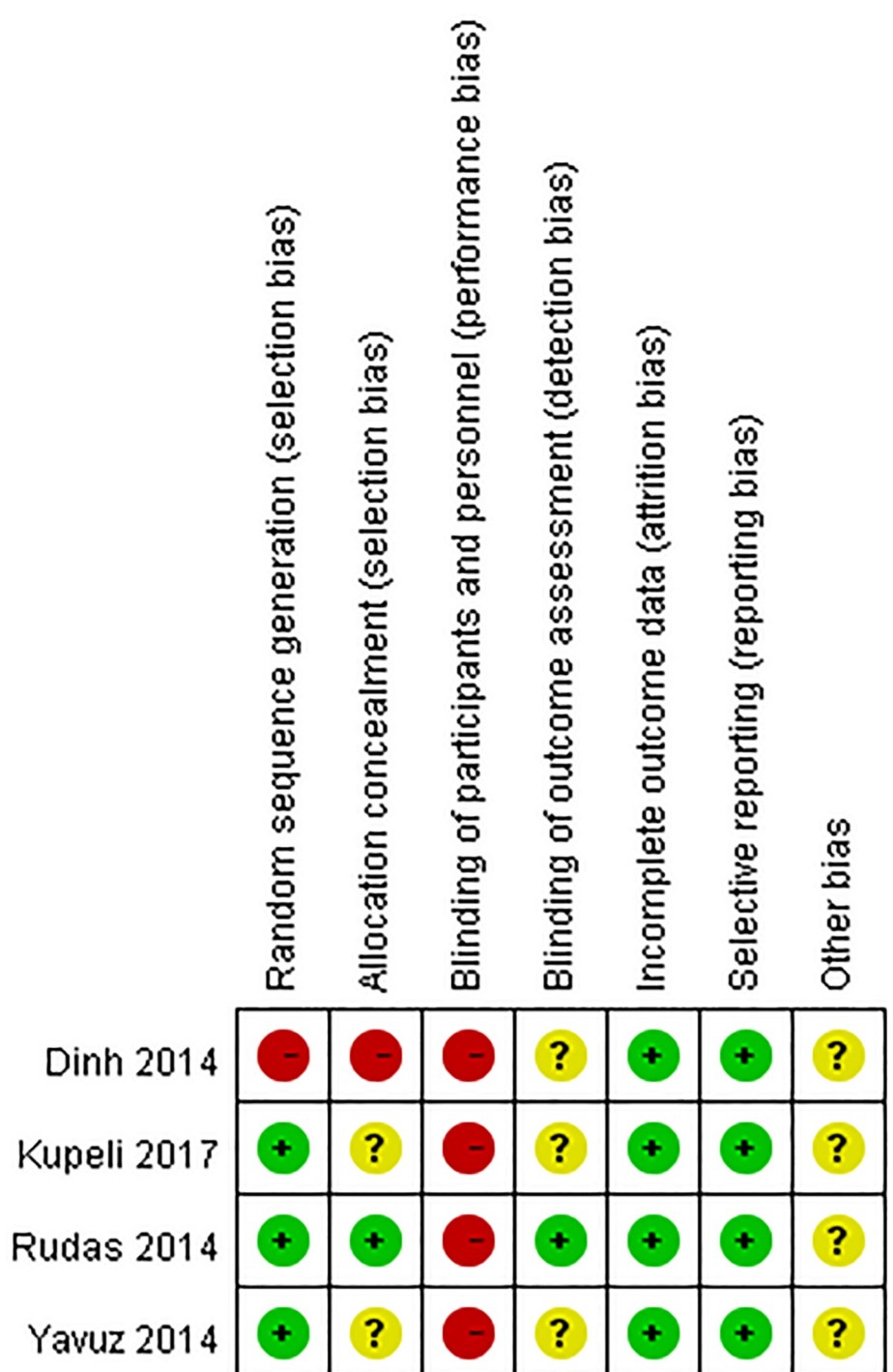

**Fig 2. Risks of bias summary.**

### Pooled ORs of first-pass success

The pooled OR of first-pass success in UGT compared with LT was 4.287 (95% confidence interval [CI]: 2.308 to 7.964) (Fig 3A). The first-pass success was significantly higher in the

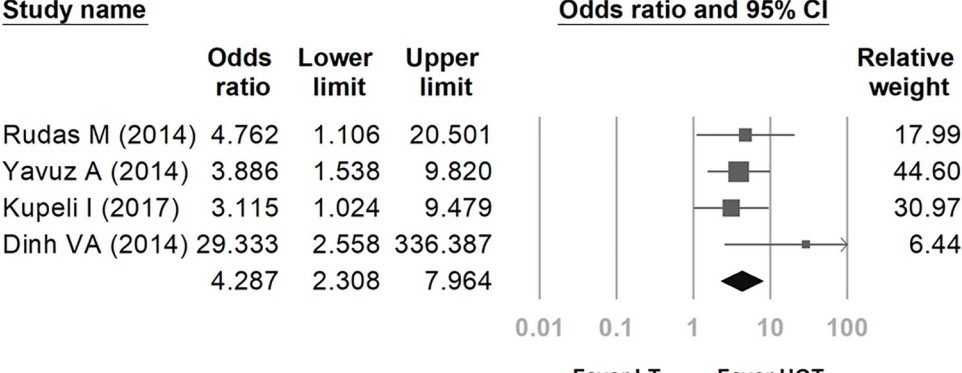

a

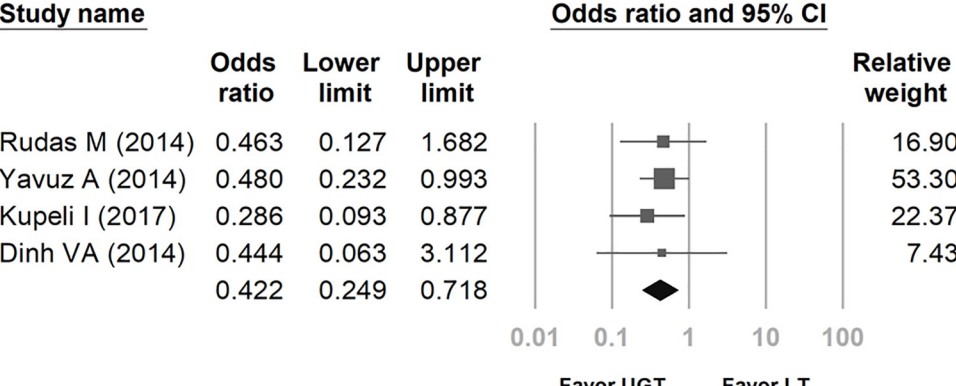

b

**Fig 3. Forest plot comparing real-time UGT with LT for first-pass success (a) and complications (b).**

UGT method. Regarding the heterogeneity of ORs, the $I^2$ test result was less than 0.01%, and the p value was 0.429.

## Pooled ORs of complications and major bleeding

The pooled OR of periprocedural complications between UGT and LT was 0.422 (95% CI: 0.249 to 0.718) (Fig 3B). The complications were significantly lower with UGT than with LT. Regarding the heterogeneity of ORs, the $I^2$ test result was less than 0.01%, and the p value was 0.895.

The pooled OR of major bleeding between UGT and LT was 0.374 (95% CI: 0.112 to 1.251) (Fig 4A). The UGT method did not significantly improve the major bleeding rate. Regarding the heterogeneity of ORs, the $I^2$ test result was less than 0.01%, and the p value was 0.97.

## SMDs of the total tracheostomy procedure time

The SMD of the total tracheostomy procedure time between real-time ultrasound-guided long-axis percutaneous tracheostomy and LT was -0.335 (95% CI: -0.842 to 0.172) (Fig 4B). Kupeli's study including three different percutaneous tracheostomy techniques was selected for comparison between the only real-time long-axis UGT method and LT in total tracheostomy time [26]. Another study included a comparison between real-time long-axis UGT and

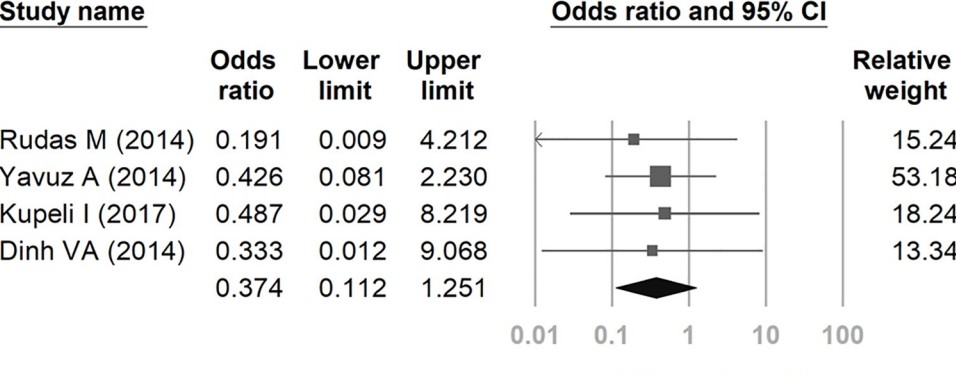

a

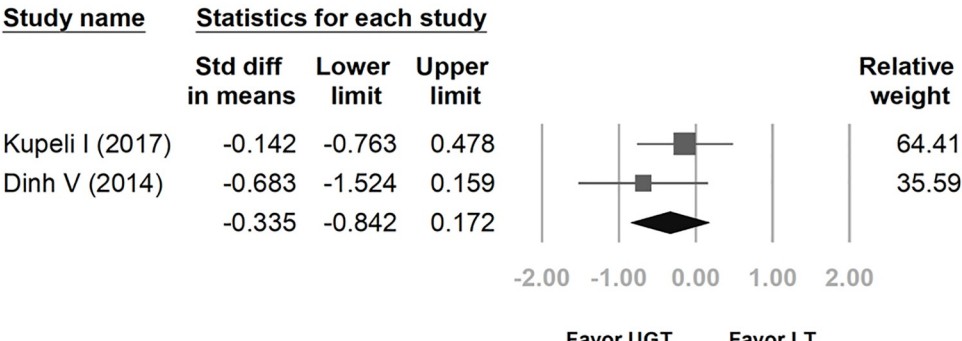

b

**Fig 4. Forest plot comparing real-time UGT with LT for major bleeding (a) and total tracheostomy procedure time (b).**

the LT method in total tracheostomy time [27]. These two publications comprised the meta-analysis. Regarding the heterogeneity of SMD, the $I^2$ test result was 2.513%, and the p value was 0.311.

The funnel plots of the log odds ratio of first-pass success, complications, and major bleeding are shown in Fig 5A–5C, respectively.

## Discussion

The present meta-analysis compared real-time UGT with LT in terms of the first-pass success, complication rate, major bleeding rate, and total tracheostomy procedure time. Real-time UGT can increase first-pass success and decrease procedure complications. However, UGT was not associated with a significant reduction in the major bleeding rate. The comparison of total tracheostomy placement time between UGT and LT was inconclusive.

Ultrasound in the short-axis plane can identify anatomic structures and the midline plane. The ultrasound long-axis plane can identify the tracheal puncture level and can trace the needle path [13]. The meta-analysis of the included studies mostly combined short-axis and long-axis scanning techniques when using real-time UGT [24, 25]. In the Kupeli study, ultrasound-guided long-axis and short-axis techniques were separately analyzed [26]. As our meta-analysis compared real-time UGT with the LT method, the separate long-axis and short-axis results in

UGT were combined for meta-analysis of first-pass success, complications, and major bleeding [26].

In Chacko et al.'s study, the first-pass success of real-time UGT was 96.8% [28]. In a review by Alansari et al., UGT reduced the number of puncture attempts [5]. In a review by Rudas et al., real-time UGT enabled clear visualization of anatomical landmarks and resulted in high success [12]. In our meta-analysis, first-pass success was higher in UGT.

In previous studies, the complication rate of UGT was 20.7% [15]. Real-time UGT was associated with a significant reduction in procedure-related complications in previous studies [11, 12, 17, 29]. In a systematic review, UGT seemed to reduce minor complications compared with LT [16]. Our meta-analysis also showed that UGT was associated with a significant reduction in all-cause complications.

In most patients receiving PDT, bleeding complications were minor, and manual compression was enough to stop the bleeding. The percutaneous tracheostomy-related bleeding rate was 4.8% in a retrospective review [30]. In a systematic review, the lethal complication rate was 0.17%, and the main cause of death in PDT was hemorrhage [31]. Ultrasound can identify the tracheal ring and vascular anatomy, therefore avoiding vessel puncture and diminishing major bleeding risks [5, 30]. However, in our meta-analysis, UGT did not significantly decrease major bleeding occurrence compared with LT.

The total tracheostomy placement time, recorded from the beginning of needle puncture to the completion of tracheostomy tube placement, was not significantly reduced in UGT compared with LT in the SMD meta-analysis. Yavuz's study in our meta-analysis recorded that the total tracheostomy placement time from the beginning of ultrasound scanning in UGT or physical palpations in LT, not from the beginning of needle puncture, was waived from the SMD calculation [25]. Since the total tracheostomy placement time was considered in only two articles, it was difficult to conclude whether UGT could reduce the total tracheostomy placement time compared with LT.

In previous studies, the median tracheostomy procedure time in UGT was 8 to 12 minutes [28, 32]. In a retrospective study, real-time UGT reduced the number of puncture attempts and shortened the operation time [20]. However, in a randomized controlled trial, real-time UGT did not shorten the total tracheostomy operation time [15]. UGT is operator dependent and associated with a long learning curve, and a well-trained physician shortened the placement time after at least 50 procedures [33]. In our meta-analysis, the total tracheostomy operation time comparison between real-time UGT and LT was inconclusive. Further study can make comparisons after matching different interventional physicians.

There were several limitations in this meta-analysis. First, the patient number and included studies were small. This small patient number made our results evidence to be less convinced. Second, the major bleeding condition was hard to define. We combined the number of major bleeding events in the recorded studies [25–27] and bleeding that required intervention as a substitute for the major bleeding events [24]. Further studies should analyze the major bleeding complications quantitatively.

## Conclusions

The present meta-analysis revealed that compared with LT, real-time UGT increases first-pass success and decreases complications. However, UGT was not associated with a significant reduction in the major bleeding rate. The total tracheostomy placement time comparison between UGT and LT should be further investigated. UGT is a safe and feasible method for percutaneous tracheostomy placement.

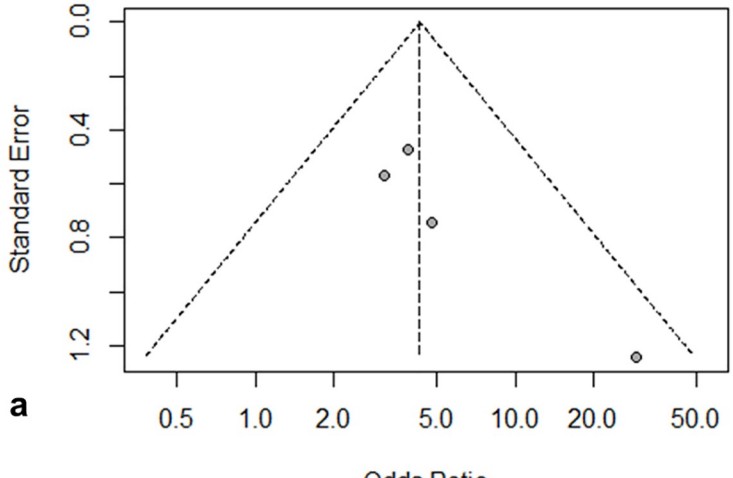

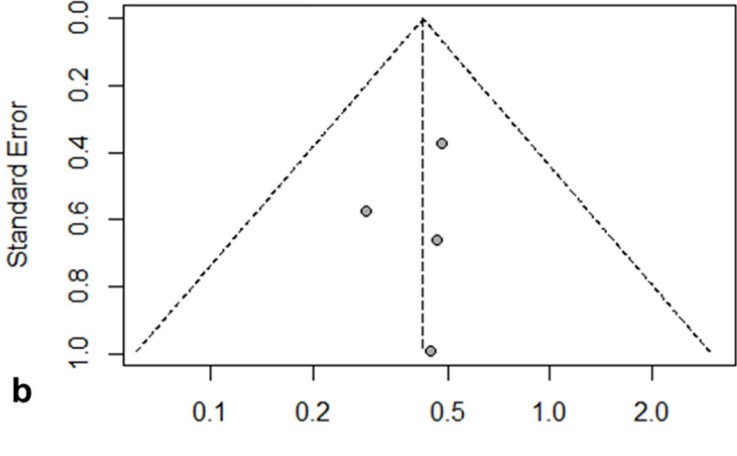

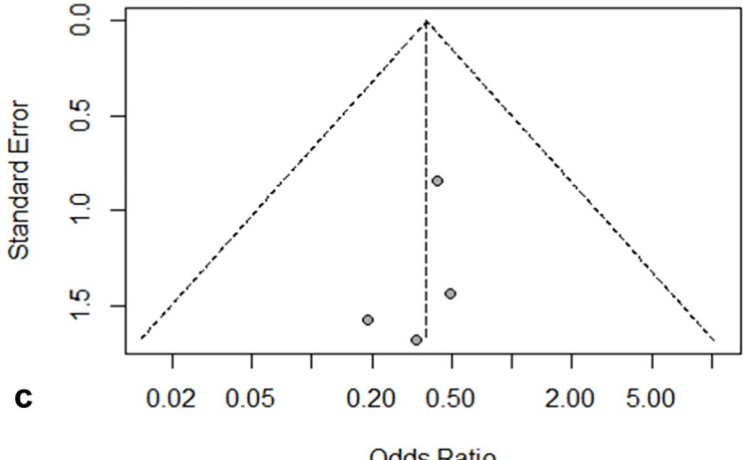

**Fig 5. Funnel plots of first-pass success (a), complications (b), and major bleeding (c).**

## Supporting information

**S1 Checklist.**
(PDF)

**S2 Checklist.**
(PDF)

## Acknowledgments

All authors express the greatest appreciation to Mrs. Mei-Chueh Yang (a research assistant at Changhua Christian Hospital, Changhua, Taiwan) for her support in data retrieval and the writing of this manuscript.

## Author Contributions

**Conceptualization:** Kun-Te Lin.

**Data curation:** Chun-Wen Chiu, Chi-Hsien Lin.

**Formal analysis:** Chun-Wen Chiu, Chi-Hsien Lin.

**Methodology:** Yung-Shuo Kao.

**Supervision:** Chu-Chung Chou, Yan-Ren Lin.

**Writing – original draft:** Kun-Te Lin.

**Writing – review & editing:** Pei-You Hsieh, Yan-Ren Lin.

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
