## [Decision Letter · Decision Letter 0]

26 Jul 2021

PONE-D-21-11232

Comparative effectiveness of ultrasound-guided and anatomic-landmark percutaneous dilatational tracheostomy: a systematic review and meta-analysis

PLOS ONE

Dear Dr. Yan-Ren,

Thank you for submitting your manuscript to PLOS ONE. After careful consideration, we feel that it has merit but does not fully meet PLOS ONE’s publication criteria as it currently stands. Therefore, we invite you to submit a revised version of the manuscript that addresses the points raised during the review process.

We look forward to receiving your revised manuscript.

Kind regards,

Oded Cohen

Academic Editor

PLOS ONE

“NO”

d) If you did not receive any funding for this study, please state: “The authors received no specific funding for this work.

Additional Editor Comments (if provided):

The authors have done good job in their manuscript, as reflected in the reviewers' comments. I agree that for this paper to be published, a thorough English editing is needed, by a professional English Editor.

Reviewers' comments:

Reviewer's Responses to Questions

**Comments to the Author**

1. Is the manuscript technically sound, and do the data support the conclusions?

Reviewer #1: Yes

Reviewer #2: Yes

2. Has the statistical analysis been performed appropriately and rigorously? 

Reviewer #1: Yes

Reviewer #2: Yes

3. Have the authors made all data underlying the findings in their manuscript fully available?

Reviewer #1: Yes

Reviewer #2: Yes

4. Is the manuscript presented in an intelligible fashion and written in standard English?

Reviewer #1: Yes

Reviewer #2: Yes

5. Review Comments to the Author

Reviewer #1: Dear editor and authors- thanks you for considering me to review this paper.

This work thoroughly covers the knowledge regarding UPT vs. LT

a few minor issues to be considered:

1. safety of BPT - in the introduction there is some concern regarding high ICP and high AW pressure with BPT. the BPT safety has been reported in other trials. regarding the high ICP- bronchoscopy is done in TBI patients for other indications as well and it is not clear from the reference if that is a consequence of insufficient sedation . as for high AW pressures - again- the high AW pressure in BPT is secondary to high resistance and so the relation to PNX is loose.

2. total tracheostomy placement time is considered in only 2 papers (the third was measured differently) and so the data are limited to conclude that UPT is not associated with time reduction, or the other way around .

Reviewer #2: This study is a systematic review and meta analysis comparing ultrasound guided percutaenous dilatation tracheostomy compared to anatomical landmark percutaenous dilatation tracheostomy.

The study was well planned and well executed.

There are several minor revisions I would suggest:

-While the English in the study is intelligible, the manuscript, and especially the discussion section would benefit from linguistic editting.

- Abstract :

Numbered references are not used in abstracts. Please revise.

- Results:

The font in figure 5 is too small, and it is lacking headlines.

6. PLOS authors have the option to publish the peer review history of their article (what does this mean?). If published, this will include your full peer review and any attached files.

Reviewer #1: **Yes: **Stavi, Dekel

Reviewer #2: **Yes: **Yael Shapira-Galitz

---

## [Author Response · Author response to Decision Letter 0]

10 Aug 2021

Dear Editor:

 Enclosed is our revised manuscript (Re: PONE-D-21-11232 Comparative effectiveness of ultrasound-guided and anatomic landmark percutaneous dilatational tracheostomy: A systematic review and meta-analysis). We appreciate your constructive comments, and we have made revisions accordingly.

Please find our responses to your comments below.

1. The authors have done good job in their manuscript, as reflected in the reviewers' comments. I agree that for this paper to be published, a thorough English editing is needed, by a professional English Editor

Answer: We appreciate your comments. This manuscript has been revised by a professional English editing service (https://www.aje.com).

 

Dear Reviewer #1 :

 Enclosed is our revised manuscript (Re: PONE-D-21-11232 Comparative effectiveness of ultrasound-guided and anatomic landmark percutaneous dilatational tracheostomy: a systematic review and meta-analysis). We appreciate your constructive comments, and we have made revisions accordingly.

Please find our responses to your comments below.

1. Safety of BPT - in the introduction there is some concern regarding high ICP and high AW pressure with BPT. the BPT safety has been reported in other trials. regarding the high ICP- bronchoscopy is done in TBI patients for other indications as well and it is not clear from the reference if that is a consequence of insufficient sedation. as for high AW pressures - again- the high AW pressure in BPT is secondary to high resistance and so the relation to PNX is loose.

Answer: We appreciate your comments. We added reference No. 6 to explain that the benefit of decreasing complications was controversial under BGT in the “Introduction” section. The previous sentence that BGT was not suitable for traumatic brain injury patients has been removed. We also edited the explanation for BGT-induced high airway pressure and removed the original pneumothorax complications in the revised manuscript.

On page 4 line 9: BGT provides benefits in the real-time confirmation of needle placement, midline positioning of the needle, and avoidance of posterior tracheal wall injury [4,5]. However, the benefit of reducing the complication rate is not significantly observed [6,7]. A previous study also noted that increased airway resistance during BGT would secondary increase high airway pressure [8]. Therefore, considering patient safety and cost effectiveness, BGT is challenging among PDT procedures [7].

2. Total tracheostomy placement time is considered in only 2 papers (the third was measured differently) and so the data are limited to conclude that UPT is not associated with time reduction, or the other way around.

Answer: We agree with your comments. Considering that the total tracheostomy placement time was considered in only two papers, we modified the conclusion that the comparison of total tracheostomy placement time between UGT and LT should be further investigated.

On Page 3 line 3: The total tracheostomy placement time comparison between UGI and LT was inconclusive.

On Page 23 line 1: Since the total tracheostomy placement time was considered in only two articles, it was difficult to conclude whether UGT could reduce total tracheostomy placement time compared with LT. 

On Page 24 line 6: The total tracheostomy placement time between UGT and LT should be further investigated.

 

Dear Reviewer #2:

 Enclosed is our revised manuscript (Re: PONE-D-21-11232 Comparative effectiveness of ultrasound-guided and anatomic landmark percutaneous dilatational tracheostomy: a systematic review and meta-analysis). We appreciate your constructive comments, and we have made revisions accordingly.

Please find our responses to your comments below.

1.-While the English in the study is intelligible, the manuscript, and especially the discussion section would benefit from linguistic editing.

Answer: We appreciate your comments. We have revised the manuscript, especially the “Discussion” section, for coherence. Additionally, this manuscript has been revised by a professional English editing service (https://www.aje.com).

2.- Abstract: Numbered references are not used in abstracts. Please revise.

- Results: The font in figure 5 is too small, and it is lacking headlines.

Answer: We appreciated your suggestions. The numbered references were deleted from the “Abstract,” and Figure 5 was modified for clarity in the revised manuscript. Please see the Abstract and Figure 5.

---

## [Decision Letter · Decision Letter 1]

11 Oct 2021

Comparative effectiveness of ultrasound-guided and anatomic-landmark percutaneous dilatational tracheostomy: a systematic review and meta-analysis

PONE-D-21-11232R1

Dear Dr. Yan-Ren,

We’re pleased to inform you that your manuscript has been judged scientifically suitable for publication and will be formally accepted for publication once it meets all outstanding technical requirements.

Kind regards,

Oded Cohen

Academic Editor

PLOS ONE

Additional Editor Comments (optional):

I wish to commend the authors for a very well written paper

Reviewers' comments:

Reviewer's Responses to Questions

**Comments to the Author**

1. If the authors have adequately addressed your comments raised in a previous round of review and you feel that this manuscript is now acceptable for publication, you may indicate that here to bypass the “Comments to the Author” section, enter your conflict of interest statement in the “Confidential to Editor” section, and submit your "Accept" recommendation.

Reviewer #2: All comments have been addressed

2. Is the manuscript technically sound, and do the data support the conclusions?

Reviewer #2: Yes

3. Has the statistical analysis been performed appropriately and rigorously? 

Reviewer #2: Yes

4. Have the authors made all data underlying the findings in their manuscript fully available?

Reviewer #2: Yes

5. Is the manuscript presented in an intelligible fashion and written in standard English?

Reviewer #2: Yes

6. Review Comments to the Author

Reviewer #2: (No Response)

7. PLOS authors have the option to publish the peer review history of their article (what does this mean?). If published, this will include your full peer review and any attached files.

Reviewer #2: **Yes: **Yael Shapira-Galitz

---

## [Editor Report · Acceptance letter]

19 Oct 2021

PONE-D-21-11232R1 

Comparative effectiveness of ultrasound-guided and anatomic landmark percutaneous dilatational tracheostomy: A systematic review and meta-analysis 

Dear Dr. Lin:

I'm pleased to inform you that your manuscript has been deemed suitable for publication in PLOS ONE. Congratulations! Your manuscript is now with our production department. 

Kind regards, 

on behalf of

Dr. Oded Cohen 

Academic Editor

PLOS ONE